# Logogram VR: Treadmill-Coupled VR with Word Reflective Content for Embodied Logogram Learning

Gwangbin Kim [1] , Eunsol An [1] and SeungJun Kim [1,2,*]

1 School of Integrated Technology, Gwangju Institute of Science and Technology, Gwangju 61005, Republic of Korea
2 AI Graduate School, Gwangju Institute of Science and Technology, Gwangju 61005, Republic of Korea
* Correspondence: seungjun@gist.ac.kr; Tel.: +82-62-715-5352

**Abstract:** A logogram is a type of writing system in which each character represents a word. Compared to segmental scripts where the alphabets reflect sounds, learning logograms are disengaging, since each character is not linked to its pronunciation. This paper presents Logogram VR, a virtual reality edutainment game that uses a treadmill and controllers to teach Hanja, which uses logograms. Hanja is a traditional Korean language writing system comprising over 8000 Chinese characters. The system leverages the logogram's feature that each letter stands for each vocabulary item, as an embodied learning strategy. Specifically, it incorporates each character's meaning into the VR learning environment, accompanied by gamified actions using a treadmill and VR controllers. We evaluated the system with 33 participants to test its overall usability, while determining the desirable playtime and number of characters for the further enhancement of it. We demonstrated and assessed the system with 125 visitors at an exhibition to disseminate it and verify the results with a wider population sample. The user studies revealed that the system provides a playful experience for learning Hanja without severe motion sickness. The differences in age groups showed that the embodiment approach utilizing meanings and actions in VR may be an effective logogram edutainment strategy, particularly among adolescents.

**Keywords:** embodied learning; language edutainment; logogram; virtual and augmented reality

## 1. Introduction

Logograms are written language systems in which each character corresponds to a word. Although they generally have a longer history than phonograms do, logograms still play important roles in modern communication [1]. Currently, one fifth of the world's population uses logograms, mostly in east Asian countries. Speakers of Mandarin Chinese, which uses logograms in written communication, comprise the second most common language (L2) speakers, following English speakers [2]. Despite their prevalence, logograms are often perceived to be difficult to learn as a large number of characters are used [3,4]. Learning logograms requires the memorization of thousands of distinctive characters, unlike segmental scripts where the alphabets represent sounds.

The challenge of logogram acquisition is that the phonological structure and characters are unrelated. For instance, many written Chinese characters sound identical; this language system contains almost 8000 distinct words. The complexity of Chinese characters presents an additional obstacle to understanding them [5]. Indeed, reading and writing Chinese are viewed as more challenging than listening to and speaking Chinese [6].

Although previous research on VR language education, including phonetic Chinese, has been successful in immersing and engaging students, these findings are not immediately transferable to logograms as they focused primarily on conversational Chinese or the Mandarin rendering of Chinese characters [7,8]. There is a need for a different approach that incorporates the distinctive properties of logographic systems. Specifically, when one is

creating the education system, the complexity and relevance of the graphic configurations of the characters and the absence of a sound–script correlation should be considered.

In this study, we offer Logogram VR, an immersive virtual reality platform for learning logographic languages. As their meanings are replicated in the virtual reality environment, hundreds of distinct characters can add variety to the learning scenario. In addition, the system incorporates walking and handwriting gestures, which are vital to the acquisition of the Chinese language [9,10]. Furthermore, students can practice the listening and speaking components of logograms. To complete the scenario, they must articulate the meaning and pronunciation of each character after having finished the stroke. The participants indicated that the system's embodiment method is usable and provides a highly positive learning experience. The technique is extensible to a wide variety of Hanja and other countries' adaptations of Chinese characters.

## 2. Related Works

### 2.1. Virtual, Augmented Reality for Language Learning

Virtual and augmented reality (VAR) technologies are raising attention as alternatives to language learning. This is mainly because immersive technologies take the learners into a digitally created world, particularly in terms of the visual perception of learning [11], while immersion is a proven method for learning a second language [12]. The virtual reality environment also enables real-time feedback and interaction with the learning content, including virtual agents that guide the learning session. Consequently, the role of virtual and augmented reality technologies for language learning applications have been widely investigated (e.g., for English [13], Chinese [7], Spanish [14], French [15], German [16], and Japanese [17]).

Another strength of VAR language learning is its potential to build various learning environments with computer graphics. This characteristic of VAR language learning is particularly synergistic with logograms. Logograms comprise thousands of distinct characters that must be learned, which we can turn into a variety of learning content. In this paper, we provide a solution for the biggest challenge to logogram learning: vocabulary acquisition. Our system translates logograms into a virtual reality learning environment, allowing diverse and episodic learning content with character-specific media scenarios.

Most of the previous applications for language learning have focused on phonetic languages, which rely on the phonological properties of the languages. However, learning thousands of logographic characters strongly engages the visual memory, particularly the long-term memory [18]. Thus, we require a more specialized approach to learn logograms in VAR. Embodied virtual reality provides one such approach as gestures can aid information retrieval from the long-term memory [9]. Contemporary VAR offers an immersive experience beyond head-mounted displays. The real-time and accurate tracking of human positions, movements, and gestures engages people with the VR environment. Additionally, the human's physical activities form or affect the media scenario, which is called embodied interaction. Our system leverages embodiment to provide a greater sense of immersion and engagement with vivid imagery during language acquisition.

### 2.2. Embodied Learning in Language Acquisition

Embodied learning refers to a pedagogical approach to relate our physical actions such as a body movements or interactions with the environment with education and learning [19,20]. It is based upon the embodied cognition paradigm, which describes how our body and environment are related to cognitive processes [21]. Theories of embodied cognition suggest that our mind is integrated into the sensorimotor systems, influencing the way we think, infer, and conceive abstract cognitive processes [21,22]. Embodied learning is effective as body movements facilitate the retrieval of mental or lexical items as a "cross-modal prime" [9]. For these reasons, the concept of embodied learning applies to various domains (e.g., enhancing the episodic memory of elderly people [23] and learning complex concepts in mathematics [24,25], physics [26], and astronomy [27]).

Embodied learning is the most widely adopted method in language education across various languages (e.g., English [28], Chinese [29], Spanish [30], French [31], German [32], and Japanese [33], etc.). This is mainly due to the link between perception and motion during conscious linguistic processes, which is supported by many empirical studies on language comprehension [34,35], second language learning [19,36], and memory [37,38]. Indeed, a word should be represented by a sensory network that represents all of the learned experiences related to the concept in our brain [39], and children's language acquisition accompanies a multitude of sensorimotor acts [22].

Thus, among the various aspects of language learning, vocabulary acquisition is particularly effective when it is combined with relevant physical activities. Students have showed enhanced memory retrieval with an higher number of expressive words [28]. Embodied learning also improves child learners' emotional engagement while they are learning L2 vocabularies [18]. Given that the most challenging part of logogram learning is learning the extensive amount of characters, embodied virtual reality can provide aid. The effectiveness of the embodied approach in vocabulary acquisition can help in learning thousands of characters, each of which represents a morpheme. Therefore, a virtual reality application that integrates the logograms' own characteristics into the embodied learning context is required. Our system uses user's bodily gestures and speech during the embodied learning process to control the VR scenario content, reflecting the meaning of the logogram that is being learned.

### 2.3. Contribution Statement

Our approach brings the following research and practical contributions to the field:

- We introduced the feature of logograms, where a word represents a meaning, as a language learning strategy by incorporating them into the media content of the virtual reality environment.
- We introduced physical actions such as walking and gestural writing as a means of the embodiment of language learning.
- Our results demonstrate that embodied learning can provide an enjoyable experience for learning logograms, particularly for adolescents.
- We have identified the ideal playtime and number of characters and reported user feedback for the future development of virtual reality embodied learning of logograms.

These contributions are relevant to the areas of immersive learning technologies [40,41] that are discussed in the AR/MR/VR [42–45] and human–computer interaction [46–48] communities, as it involves the introduction of a virtual reality approach for logogram learning. Precisely, the research is aligned with the field of embodied learning [49,50] as it involves bodily gestures using arms and legs and speeches to facilitate lexical item retrieval and procedural memories as a cross-modal prime. It also relates to VR locomotion technology [51–53] since it utilizes a treadmill as a form of the embodiment of language learning.

### 3. Methods

This paper suggests Logogram VR, a logogram learning program that uses embodied virtual reality media. In particular, the system is designed to support the education of Hanja, which refers to the Korean version of Chinese characters. Hanja is a writing system that is used alongside the Korean alphabet (Hangeul) in Korea, and it is taught as a second language to students as part of the education. However, Hanja learning falls between first language (L1) acquisition and second language (L2) acquisition for Korean people, as it shares the same pronunciation as the Korean language, but it uses Chinese characters.

The platform's purpose is to teach the strokes, meaning, and pronunciation of each character. We designed our system based on the taxonomy for educational embodiment [27]. Each scenario reflects the meaning of each character in an environment, while it also enabling gesture- and walking-based interaction using a treadmill and VR controller. This

section describes, in detail, the hardware and software specifications, as well as the user scenario and the user studies.

Specifically, our user study aims to answer the following research questions:

- RQ1: Is Logogram VR usable enough for first-time users?
- RQ2: What are the most desirable playtime and the number of characters for each session?
- RQ3: Does the embodied learning approach support an enjoyable, engaging, immersive, and helpful logogram learning experience?
- RQ4: Which subsection of the population can benefit from the approach the most?

Usability is important to make a system apparent to users [54], preventing adoption hesitancy. This is particularly true when the system is novel [55] and targeted to non-expert end-users [56]. To answer RQ1, the strengths, weaknesses, and risk factors of the system's usability were identified, along with the overall usability in terms of a score. We set RQ1 as our primary question because system usability that prevents adoption hesitancy should be first checked when a system is new.

RQ2 was used to determine the desirable playtime and characters per session to ensure that the users are engaged with the learning content. Maintaining the users' engagement is essential considering its relationship with achievement [57]. RQ2 is related to the memory capacity that can be used for one session, and it is especially important because the system targets teenagers whose attentional control is still developing [58].

Although usability in accepting a new product is important, researchers should also think of user experience as well as usability. Indeed, interactive, fun, and enjoyable experiences affect the learning result [59–61]. RQ3 was set based upon our consideration of a representative user experience with the learning content, which is related to success in education, enjoyment [17,62], engagement [17,63], immersion [12,62], and helpfulness [62].

RQ4 addresses the difference between the age groups, i.e., teenagers and older adults, while we answered RQ1-3. We studied the difference between the age groups, focusing on teenagers, since most L2 learning occurs at this age. Additionally, despite some debates, the 'critical period' theory suggest that childhood is critical for successful L2 learning [64,65]. The younger generations' willingness to accept new technologies (e.g., "digital natives") and the limited cognitive capabilities that children have may also be related to this issue [66].

### 3.1. Implementation

#### 3.1.1. Hardware Configuration

Logogram VR consists of ⓐ a head-mounted display (HMD) that creates a VR environment, ⓑ a computer that runs the VR software, ⓒ a treadmill that detects the user's walking, and ⓓ VR controllers, with which the user can write letters (see Figure 1A for a schematic diagram and Figure 1B) for an example set-up). The VR controller enables the enlarged, gestural handwriting of logograms. The microphone built in the HMD recognizes the user's speech through the Google Speech-to-Text API with a predefined word set. The omnidirectional treadmill simulator employed in our system enables limitless movement by walking or running on the bowl-shaped platform while wearing the compatible low friction shoe covers for slippery walking. The system holds the user's waist in place safely, and it was approved by the Institutional Review Board.

#### 3.1.2. Program Flow

The system is designed to test a proof-of-concept at a design exhibition. With three characters to learn in such a densely populated environment, the demonstration scenario should last 5–10 min. The duration was set to approximately seven minutes after experiment 1, which assessed the desirable playtime and number of characters for each scenario for the participating learners.

We set Hanja, the Korean rendering of Chinese characters, as the type of logogram, considering its importance to the Korean language. Among thousands of Hanja characters, the words rain, cloud, and thunder were selected as representative scenarios for learning

(Figure 2). The character for rain belongs to a hieroglyphic logogram, whereas cloud and thunder incorporate rain in their strokes.

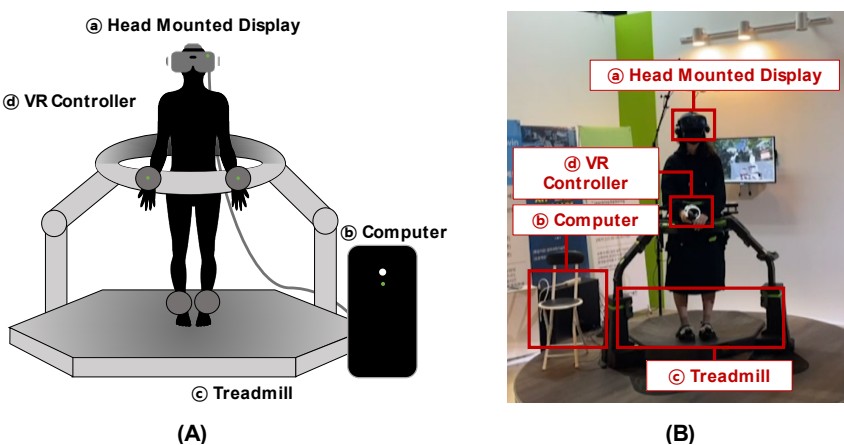

**Figure 1.** (**A**) Hardware settings; (**B**) example set-up at a design exhibition. Logogram VR comprises a head-mounted display, VR controllers, a treadmill, and a computer to execute the software.

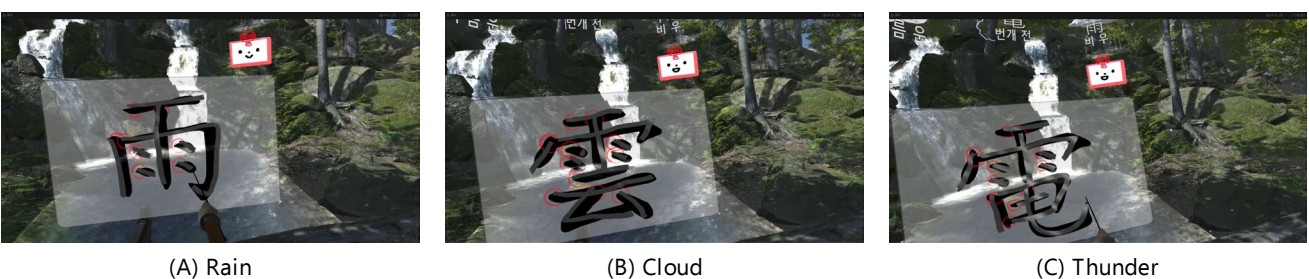

(A) Rain  (B) Cloud  (C) Thunder

**Figure 2.** Exemplary characters in the Logogram VR's user scenarios. (**A**) Rain, (**B**) cloud, and (**C**) thunder.

Figure 3 illustrates the program flow diagram. When a learner puts on the device and launches the program, the VR media begins with an environment that includes a forest and thunder. The user is free to move around within the virtual reality environment, and the virtual agent (VA) guides them to a waterfall. They are shown a brief description of the characters that they have to learn, and the virtual agent recommends that the user writes and speaks as the topic is explained. When the user finishes, the program also checks their writing and speech. Specifically, while they are writing the letter, the user can see a guide to learn the stroke sequence. When the user finishes each stroke using a VR controller, the Unity 3D colliders embedded in the guide for the strokes measure whether they achieved the correct stroke order. The speaking part is recognized by the computer using a microphone that is embedded in the HMD and Google Speech-to-Text API. The word keys were established through our initial test to guarantee recognition accuracy. Once the user's writing and speech have been successfully verified, the meaning of the letter is displayed within the media content of the virtual reality environment. For example, when the user completes the word rain, rain starts falling in the forest. These stages of the VA's explanation, the user's repetition, and the reflection in VR are repeated, and the program ends when the user completes the words in each scenario.

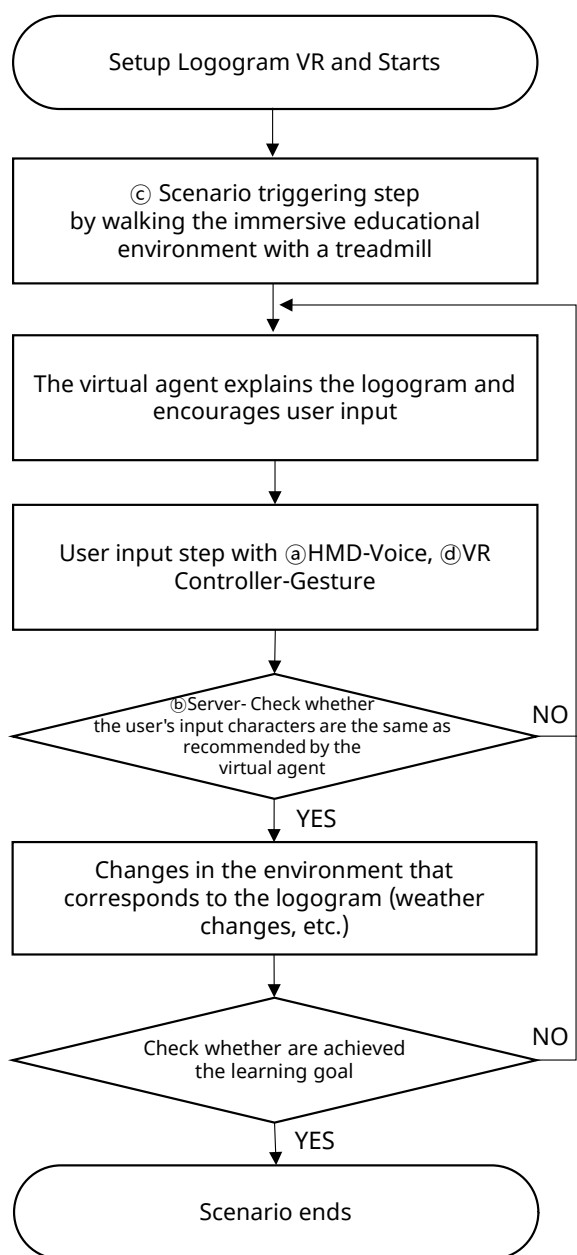

**Figure 3.** Program flow diagram.

### 3.1.3. User Scenario

In the system settings that were described earlier, the user walks through the forest to reach the waterfall (S1 of Figure 4). Subsequently, they learn the characters in the order of rain, cloud, and then thunder, considering the difficulty and composition of each word. The game's VA explains the composition principle of each character and encourages the user to write on the canvas (S2). Once the user writes a character, following the guided stroke sequences (S3), and speaks the meaning and pronunciation of the character to the waterfall (S4), the weather in the VR environment changes according to the character that has been learned (S5). After completing each character, the user can freely move and navigate the forest to experience the changing visual and auditory experiences that the weather brings (S6).

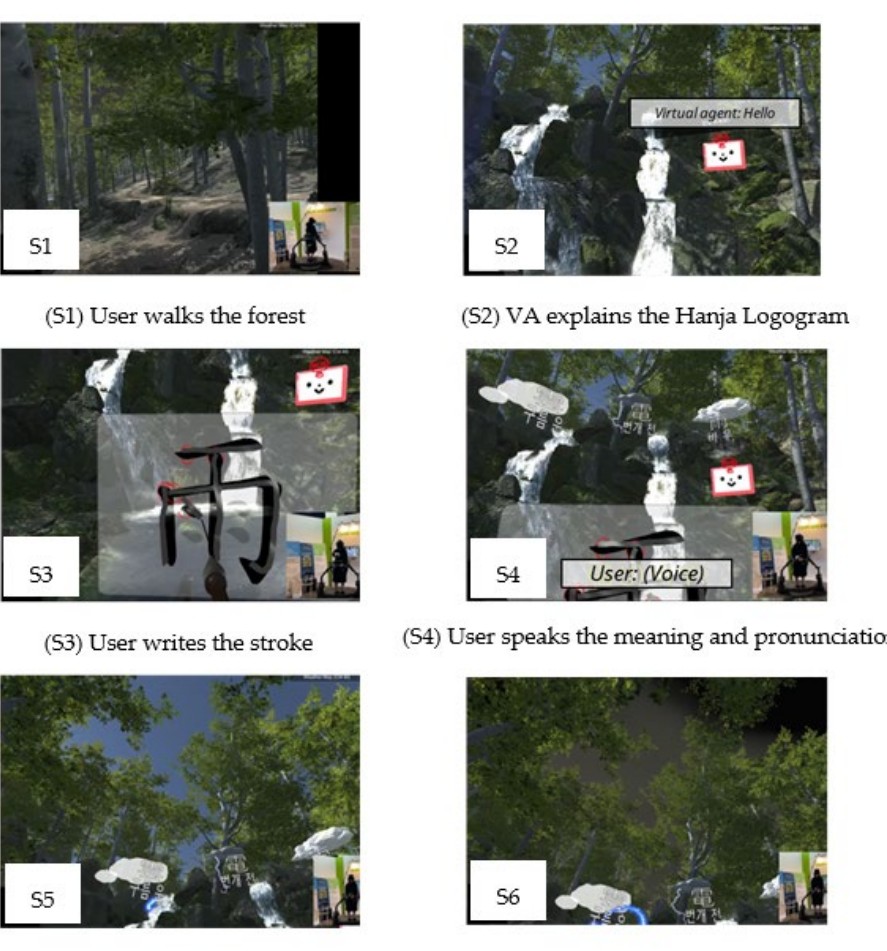

**Figure 4.** Scenes in Logogram VR. The tested scenario includes: (S1) natural locomotion, (S2) virtual agent, (S3) writing practice (Chinese character stands for 'rain'), (S4) speaking practice, (S5) content reflection, and (S6) user immersion with the word reflective content.

The gesture- and walking-based interactions intend to provides additional bodily movements and gestures that allow a "cross-modal prime" for lexical item retrieval [9] and vocabulary acquisition [19,67]. The VR locomotion-based interaction provides sensorimotor engagement, which is an essential part of embodied learning, with large areas in the sensorimotor cortex being activated [27]. We viewed the handwriting as an important embodiment component, considering a neuroimaging study [68] that discovered the relationship between the premotor cortex, a part of the sensorimotor cortex region, and handwriting and logographic character reading. In addition, the VR controller turns the gestural input of strokes into handwritings. In the virtual reality, the learner holds a large brush, and the developed gestures resemble brush calligraphy, which generates high congruence between the movements and the subsequent input metaphors. Larger muscular movements were designed to engage more sensorimotor systems, which may produce stronger learning signals [27]. Finally, we designed the user scenario to include multi-modal bodily engagement during VR learning. These activities additionally enable the efficient retrieval of procedural memories [69], which is distinct from declarative memory, to which normal L1 and L2 acquisition are related [70].

### 3.1.4. Implementation Note

We used the VIVE Cosmos for the HMD because of its built-in microphone and VR controller capabilities. The Virtuix Omni treadmill was used to simulate the user's walking because it facilitates walking in any direction [71]. The VR learning environment was run using the Unity 3D engine.

### 3.2. Experimental Method

We exhibited the system at a design biennale exhibition, which featured 400,000 total visitors for the entire exhibition. An exhibition is an environment that features possibilities for ambient sounds, bystanders, and a staging effect [72], which can be an effective space to test the use of virtual reality in public spaces such as classroom environments. Public spaces also have risks such as colliding with others, which can be avoided by physically separating people [73]. However, in an empirical study, the difference between the user behaviors in a public space and a fully separated environment was insignificant [73]. In our case, we allocated exclusive space for the treadmill and equipped it with safety bar to guarantee physical separation, assuring that the users immerse themselves into the VR environment.

We conducted two-fold experiments during the exhibition to test and substantiate the practicality of Logogram VR (see Figure 5): (1) System Usability Scale and surveys to enhance it, and (2) a test of the broader demographic populations to verify the group differences. The two experiments followed the same procedure, except for the questionnaire items, and no participant underwent both of the experiment. We assessed the overall usability of the system in the first phase of the user studies, prior to the system's dissemination. We also confirmed the number of characters per scenario and the content duration per character. In the second experiment, we verified how the public perceives the system, and as well as this, we performed a further analysis of the demographic population groups. The participants that were aged 10–19 were classified as teenagers, and people that were aged 20+ were classified as adults, which is in line with the Word Health Organization (WHO)'s classification of adolescents (10–19) and adults (20+). No participants were below 10 years old in order to follow the IRB's guidelines.

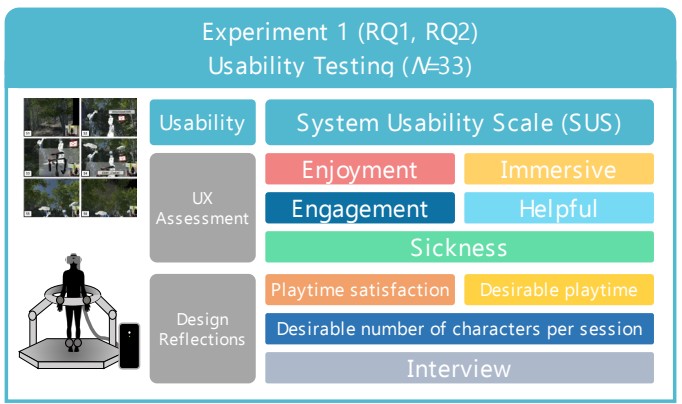
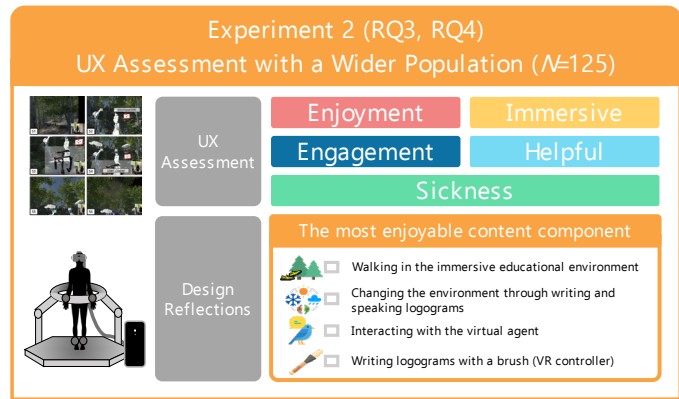

**Figure 5.** An overview of the experimental design. Experiment 1 evaluated the general usability, whereas experiment 2 tested and disseminated the users' experienced to a larger community. The UX assessment questionnaires are shared by the both of the experiments.

### 3.2.1. Experiment 1: Usability Testing (RQ1 and RQ2)

We assessed the usability of the system with 33 study participants who were visitors ($M_{age}$ = 27.24, $\sigma_{age}$ = 12.18, $Min_{age}$ = 11, $Max_{age}$ = 52, $N_{teenagers}$ = 11, and $N_{adults}$ = 22). Thirteen participants had some knowledge on at least one Chinese character that had to be learned. The participants had the experiment explained to them and answered a pre-experiment survey including demographic information. We also included questionnaire items on their prior experience or knowledge of VR, as these sometimes affect the usability

of immersive systems [74]. After experiencing the system, they answered the questionnaires, including the System Usability Scale (SUS) [75] and a custom questionnaire on enjoyment, engagement, immersion, helpfulness, and sickness to assess the users' experience (UX) using the method. The questionnaire items were selected considering the E2I scale [76] and collected on a 5-point Likert scale. SUS comprises even-numbered items with negative questions. We inverted negative questions and converted each item into a scale between 0 and 10 to sum up 100 SUS scores in total. According to Lewis and Sauro [77], SUS consists of two subscales: usability (items 1–3 and 5–9) and learnability (items 4 and 10). We averaged the 0–10 normalized items to calculate the scores for both of the subscales. The survey also asked about the desired playtime and the number of characters for a single session, and it concluded with a brief interview regarding future development ideas. The experiment lasted approximately 30 min for each participant.

### 3.2.2. Experiment 2: UX Assessment (RQ3 and RQ4)

We recruited 125 participants who were visitors ($M_{age}$ = 27.34, $\sigma_{age}$ = 10.23, $Min_{age}$ = 10, $Max_{age}$ = 60, $N_{teenagers}$ = 24, and $N_{adults}$ = 101) to assess the users' experiences among a wider population. Fifty-eight participants had some knowledge on at least one Chinese character that had to be learned. The participants had the experiment explained to them and answered a pre-experiment survey including demographic information. We also included questionnaire items on prior experience or knowledge of VR, as these sometimes affect the usability of immersive systems [74]. After experiencing the system, they answered the questionnaires, the UX assessment questionnaires (on enjoyment, immersive, engagement, helpful, and sickness). The participants were also asked to name their favorite aspect of the system to collect design reflection perspectives. The experiment, including the experience and survey, took approximately 15~20 min.

## 4. Results

This section discusses the findings of the two user studies. Some surveys were obtained solely from experiment 1, whereas the others were obtained from both of the experiments (see Figure 5 for an overview of the study design). To determine the group of the population that may benefit the most from the system, we split the participants into two groups for the analysis: teenagers and adults.

### 4.1. Usability of the System (N = 33, RQ1, and RQ4)

The system showed satisfactory usability, with an average SUS score of 75.30, where $\sigma$ = 14.15, as illustrated in Figure 6. The resulting score means that the overall usability of the system is 'good' (>71) [78] and 'acceptable' (>70) [79]. No significant difference in the overall usability between the teenagers (*M* = 75.45, $\sigma$ = 14.48) and adults (*M* = 75.22, $\sigma$ = 14.32) was found to be *t*(31) = 0.781, *p* = 0.441. There were no significant differences in the subscale usability (items 1–3, 5–9) between the teenagers (*M* = 7.386, $\sigma$ = 1.696) and adults (*M* = 7.713, $\sigma$ = 1.495), *t*(17.989) = 0.542, *p* = 0.575. Thus, the system was perceived to have 'good' and 'acceptable' usability regardless of the participants' age.

In the subscale learnability (items 4 and 10), the teenagers rated it higher (*M* = 8.182, $\sigma$ = 1.800) than the adults did (*M* = 6.761, $\sigma$ = 1.954), *t*(21.678) = 2.076, *p* = 0.049. The results show that teenagers are more convinced that they can be familiarized with the system's functions and capabilities quickly. In particular, teenagers rated it higher for item 4 ("*I think that I would need the support of a technical person to be able to use this system*") (*M* = 7.954, $\sigma$ = 0.75) than the adults did (*M* = 5.568, $\sigma$ = 1.151), which means the teenagers required less technical assistance, $U(N_{teenagers}$ = 35, $N_{adults})$ = 63.5, *z* = 2.271, *p* = 0.026. As a consequence, they were more confident in learning to use the system and using it without technical assistance than the adults were.

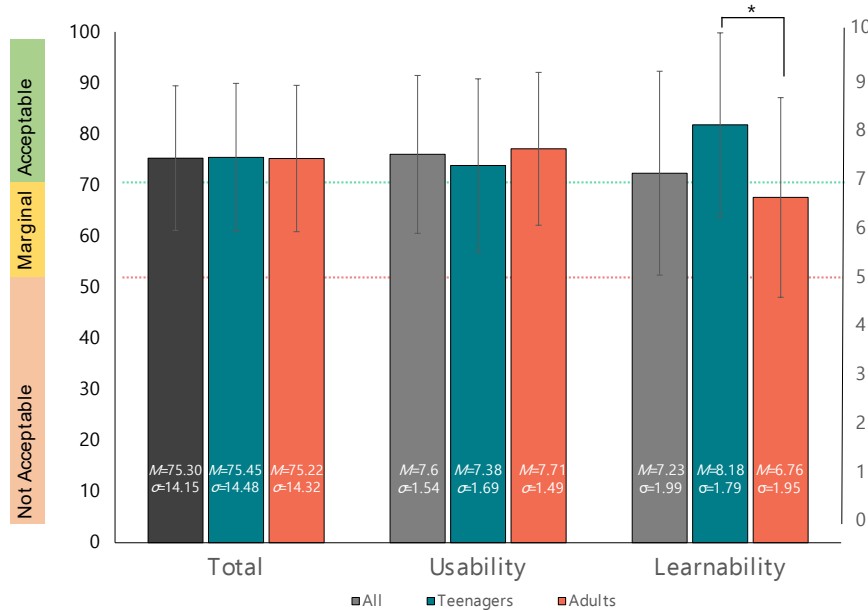

**Figure 6.** SUS score of Logogram VR (*N* = 33). Teenagers perceived Logogram VR to be significantly more learnable than adults did (* *p* < 0.05).

*4.2. Desirable Measures of Each Scenario (N = 33, RQ2)*

The participants reported that the playtime of a single episode was appropriate (*M* = 3.82, *σ* = 1.04). The average desirable playtimes were 8.91 among the teenagers and 6.05 among the adults, *t*(31) = 1.845, *p* = 0.075, while summing both of the groups resulted in *M* = 7.00, *σ* = 4.36.

For an adequate number of characters per episode, the teenagers reported *M* = 3.18, *σ* = 1.168, while the adults reported *M* = 4.41, *σ* = 3.594, but the difference did not have statistical significance, *t*(31) = 1.096, *p* = 0.281. The mean of the appropriate number of characters in both of the groups was *M* = 4.00, *σ* = 3.04.

We calculated the proper playtime per character by dividing the desirable playtime with the number of characters as per participant responses. On average, the preferable time per character was *M* = 2.17, *σ* = 1.65. The appropriate time per character was greater among the teenagers (*M* = 3.21, *σ* = 1.96) than it was for the adults (*M* = 1.65, *σ* = 1.21), with a statistical significance *t*(31) = 2.834, *p* = 0.008 (Figure 7).

*4.3. UX with the System (N = 158, RQ3, and RQ4)*

The questionnaire results show that the participants found the system enjoyable, engaging, immersive, and helpful, while it did not cause severe sickness. Figure 8 summarizes the result. The results are valid for our specific system and experiment, but it should be noted that factors such as VR exposure time can affect the severity of the sickness [80].

A Mann–Whitney U test between the participant groups revealed that the teenagers perceived the system to be significantly more enjoyable (*M* = 4.08, *σ* = 0.78) than the adults did (*M* = 3.69, *σ* = 1.02), *U*($N_{teenagers}$ = 35, $N_{adults}$) = 1698.0, *z* = 2.000, *p* = 0.046. In contrast, the teenagers reported significantly less sickness (*M* = 1.91, *σ* = 1.26) than the adults did (*M* = 2.55, *σ* = 1.25), *U*($N_{teenagers}$, $N_{adults}$) = 1477.0, *z* = 2.925, *p* = 0.003. However, there was no significant difference between the groups in terms of engagement (*U*($N_{teenagers}$ = 35, $N_{adults}$) = 1863.5, *z* = 1.252, *p* = 0.211), immersion (*U*($N_{teenagers}$ = 35, $N_{adults}$) = 2037.0, *z* = 0.500, *p* = 0.617), and helpfulness *U*($N_{teenagers}$ = 35, $N_{adults}$) = 1994.0, *z* = 0.701, *p* = 0.483). The results demonstrated that the system could be especially effective among teenagers, with higher enjoyment and less sickness.

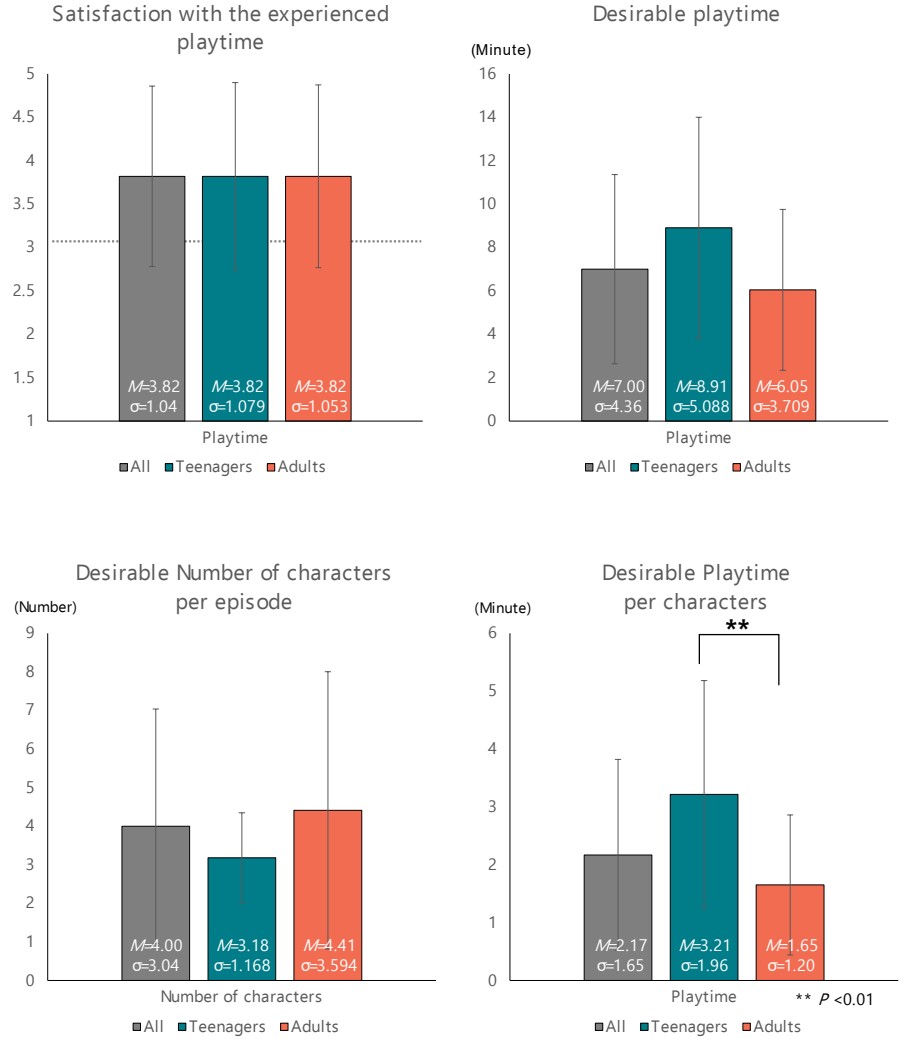

**Figure 7.** Desirable playtime and number of characters for Logogram VR (*N* = 33). Teenagers required significantly more playtime per characters than adults did (** *p* < 0.01).

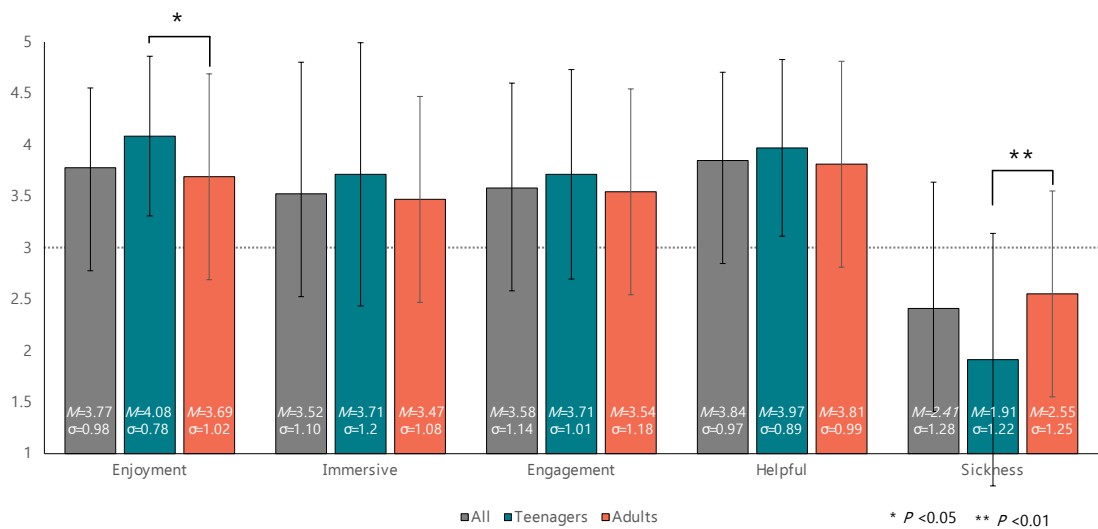

**Figure 8.** Comparison on the UX with Logogram VR between teenagers and adults (*N* = 158). Teenagers enjoyed significantly more reported significantly less sickness than adults did (* *p* < 0.05; ** *p* < 0.01).

*4.4. The Most Enjoyable Part (N = 125, RQ3)*

According to eighty percent of the respondents, the embodiment element was the most enjoyable aspect of the system. The most enjoyable part of the program was 'walking in the immersive educational environment' (76 participants, 61%), followed by 'writing logograms using a brush (VR controller)' (24 participants, 19%), as illustrated in Figure 9. The virtual agent was the least relevant part of game's enjoyable elements (7 participants, 6%).

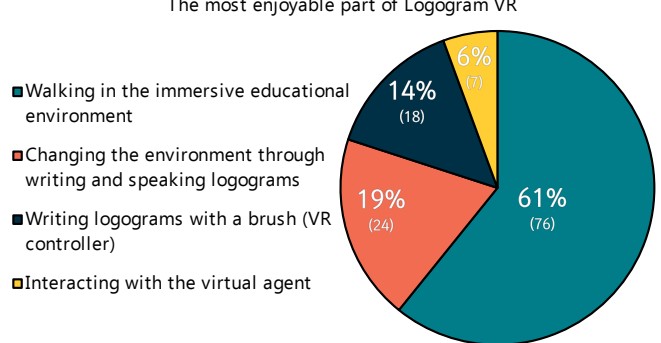

**Figure 9.** The most enjoyable elements of Logogram VR (*N* = 125). Embodiment learning components such as walking, writing, speaking were rated to be the most enjoyable parts.

## 5. Discussion

*5.1. Main Contributions*

The system employed essential elements of logograms in its process of embodied learning: graphic, semantic, and phonetic components. Graphic strokes are made with gestural inputs using virtual reality brushes. The learners also practiced the meanings and pronunciations by speaking them to the speech recognition module using the microphone in HMD. The system incorporates the logogram's unique characteristics into the learning content and environment. There being thousands of distinct characters enable a variety of different scenarios by virtue of the graphical virtual reality.

The system was assessed with a case study on Hanja, the Korean rendering of Chinese characters, but the system can be applied to other various logograms. In particular, for Chinese characters, most East Asian countries share the same logogram with different phonetic sounds (e.g., Kanji and Chu Nom). One may transplant the system across different cultures and logograms by making content adjustments and adaptations.

The user study reports reveal how the effective embodied approach might be used for learning logograms, particularly among teenagers in terms of usability and user experience. Additionally, the interview reports from the study participants provide lessons for future work, namely alternative locomotion techniques, repetitive learning, and system design for elderly people.

*5.2. Usability and User Experience of the System*

The Logogram VR was rated to have a 'good' and 'acceptable' usability of 75.30 average scores. There was no difference between the two age groups, indicating the system is usable regardless of the user's age. The participants also reported the system as being usable, enjoyable, engaging, immersive, and helpful. Above all, they responded that the embodiment actions using the hands and feet that the system provides made the learning more enjoyable (Figure 7). The participants' responses in the interviews included "*P4: I liked the gamified way of learning logograms that involved walking along with the VR, writing with an arm, and speaking out loud.*" and "*P12: It was interesting that one can learn naturally while playing the game. I think the system also helps memorize the character longer.*" The embodied learning strategy used in the study was effective in enticing the learners.

Since the experiment was conducted at a highly populated exhibition, the participants took part in the study without a training session. The positive learning experience reported

by participants is even more significant because it was obtained with them having no time to adjust. Considering that the ambient sound during the exhibition may have disrupted the participants during the gamified learning, it can be more immersive when it is used in quiet spaces.

However, a notable difference was found in the SUS item 4 ("*I think that I would need the support of a technical person to be able to use this system*"), concerning the need for technical assistance to use the system. The adults' average score for the SUS item 4 was 5.568, which slightly surpassed the neural score of 5. In contrast, teenagers did not really require technical assistance with an average score of 7.954. The other differences were found in the measures of 'enjoyment' and 'sicknesses', both of which were rated more positively among the teenagers. This could be due to the differences in the technology perceptions and adoption between the two age groups [66]. In particular, the treadmill may have affected the negative responses ("*P9: I constantly needed help because the shoes constantly came off*" and "*P11: It was hard to wear the shoes on*"). However, the result is specific to the system configuration, including the employed gamified scenario. Thus, usability and the user experience among adults can be further improved by adjusting the scenarios, and the maximum playable time without experiencing sickness should also be investigated.

### 5.3. Desirable Volumes of Each Scenario

Both the desirable playtime and the desirable number of characters per episode did not produce s statistically significant difference among the age groups. However, what is worth our attention is the individual difference. The average preferred playtime was 7.00 min, with a standard deviation of 4.36. The mean preferred number of characters was 4.00, with a standard deviation of $\sigma = 3.04$. A further statistical analysis revealed that prior knowledge and prior experience with VR, as well as age, were not contributing factors. There were no differences between the desired playtime and the number of characters per episode between the age groups, while the individual data were distributed with a large standard deviation.

The variations among the individuals in the results indicate that the playtime and the number of characters should be adaptable for a more satisfying user experience. The system could better cater to individual needs by allowing the users to decide when to move on to the next character or stop. Alternatively, the system could present groups of words based on difficulty and semantic similarity, allowing the individuals to choose whether to continue learning another group of words or to end the current session. Likewise, the lessons from previous logogram learning studies should also be applied to improve the system.

Although the desirable playtime showed a difference between the teenagers and adults, there was marginal significance in this, $t(31) = 1.845$, $p = 0.075$. A larger number of participants may produce a statistically significant result, which cannot be confirmed at this stage.

In contrast, the desirable playtime per character, which corresponds to the preferred time spent on a task, did show statistical significance. The teenagers demanded more play time ($M = 3.21$, $\sigma = 1.96$) per characters than the adults did ($M = 1.65$, $\sigma = 1.21$), $t(31) = 2.834$, $p = 0.008$. However, the group's difference between the people with prior knowledge of the characters and the others was insignificant ($U(N_{knowledgeable} = 13, N_{the\ other} = 20) = 125.5$, $z = 0.167$, $p = 0.867$). The result implies that the teenagers required more time per character than the adults did, regardless of prior knowledge. This should be interpreted in relation with discussions on 'engagement time', as the time spent on a task factor should be a more effective and efficient than a simply measuring the increase that deteriorates as the learning progresses [57].

### 5.4. Lessons for Future Development

According to Tan [81], the ease of use of an e-learning system has an impact on how useful it is perceived to be. This perception, in turn, influences both the attitude of the users towards the system, as well as their intention to use it, which eventually affects the actual

usage of the system. Thus, the usability of Logogram VR makes it a suitable system for Hanja learners, particularly adolescents. However, we also found some suggestions for the enhancement of it in the interview.

Whereas walking and interacting with the environment was mostly praised by participants, most of complaints focused on the treadmill, including two responses on the shoes in the earlier section. Indeed, the participants were fond of the walking part of the system ("*P9: Walking in the forest was the most interesting part of the game while engaged with the learning content*"). However, they raised questions on whether the treadmill is the best method of implementing such a function ("*P9: the floor was too slippery*", "*P20: P27: treadmill was not as natural as free walking*", "*the treadmill did not bring adequate sense of speed that matches the visual experience*"). There were some opinions about the need for more natural and realistic interactions for turning ("*P32: The system should enable freer change of walking orientation*"). Since walking was found to be the most enjoyable aspect of the system, removing the discomfort associated with using a treadmill will make the system more efficient and cost effective. New and efficient methods for free locomotion in VR spaces are gaining attention (e.g., walking in place [82,83] and re-directed walking [84–86], etc.), and these can be applied to this system, allowing a more affordable implementation of the system.

Some of the participants suggested learning by repetition ("*P15: The system should support repeated learning.*"). Repetition is basis of a language acquisition as it is related to the automaticity for spontaneous language production [87]. In an empirical study, better L2 performance was obtained with repetition [88]. However, it conversely affected the willingness to learn and enjoy it [88]. Similarly, in a Chinese learning classroom, it was described as being "boring" [7]. All of these aspects must be carefully considered to provide a better learning experience in future work.

Some people also commented about the motion fidelity (*P12:* "*the discrepancy between actual movement and its reflection in VR should be narrowed*"). The visual fidelity (*P5 and P29:* "*graphical realism*") and display resolution ("*P4: the display needs to be improved to be more realistic*") were also mentioned during the interview. These factors should be designed in harmony with each other to prevent severe motion sickness.

*5.5. Needs for More Extensive Investigations*

We designed our custom questionnaires referring to established questionnaires (e.g., the Presence Questionnaire [89], the E2I scale [76], and the Virtual Reality Sickness Questionnaire [90], etc.) in extant research. The entire questionnaire was reduced to simple items to capture the immediate reactions from people of various ages ($Min_{age}$ = 10, $Max_{age}$ = 60), since the desirable playtime is short (approximately seven minutes). However, questionnaires with a higher number of items will further validate the findings.

The primary target of the system is teenagers, since most of the first and second language acquisition happens during that particular period. Yet, language education is also an important part of lifelong education, and an increasing number of elderly people are participating in L2 learning. The study will guide lessons for embodied logogram learning strategies for elderly people when a greater number of older adults is involved. In the current stage, we could only involve three older adults aged 60+, leaving an insufficient amount of survey data for the statistical analysis. Considering the gap in familiarity with VR media and the hesitancy of adopting new technologies, elderly people may report different results from those of teenagers and younger adults. Given the aging-related decline in memory and other cognitive capabilities that elderly people might encounter, the experience of older people with the embodied learning strategy should also be investigated in the future.

The virtual agent was found to be the least relevant element in terms of the enjoyment that Logogram VR provides, with only 6% of the participants in experiment 2 responding that it was the most enjoyable aspect. Depending on the user preferences, the function

of the virtual agent could potentially be replaced with more conventional instructional approaches, such as written or spoken commands.

### 5.6. Limitations and Future Work

The overall usability and user experience provided by the system were the focuses of our research, as this was a first attempt to provide embodied logogram learning to the public. We verified that the embodied virtual reality approach brings an enjoyable, immersive, engaging, and helpful logogram learning experience. The usability and learning experience of an educational system is important for a successful learning [17,62]. Empirical studies show that interactivity, fun, and enjoyment positively affect the learning results [59–61] by inducing greater retention [60].

However, for an educational system, the learning that a system supports should also be investigated [74]. Indeed, 2/3 of the VR language learning research include learning gain-related metrics [91]. The current study has shown that an embodied VR approach can be an effective way to provide enjoyable learning experiences for teenagers. Future research should conduct a long-term experiment to evaluate the learning outcome with the detailed assessment and classification of the knowledge of the learner group in order to further validate the effectiveness of the system. This could be achieved by comparing the results to those results obtained from traditional methods such as books, tutors, or other e-learning methods.

We set the Chinese characters for 'cloud', 'rain', and 'thunder' as the learning content in our user studies. The system successfully integrated the meaning of the chosen characters in the VR environment. However, it is likely that additional explanations or more complex VR scenarios are required for learning abstract words. This may influence the time spent on a task that, which user attention and engagement. The usability of and user experience with the system when they are learning logograms for abstract concepts should also be investigated to guarantee the widespread adoption of the system.

### 6. Conclusions

In this paper, we introduced Logogram VR, an embodied virtual reality edutainment system for learning Hanja, the Korean rendering of Chinese characters, which is a form of logographic writing. The VR system employs an embodied learning approach by demonstrating the meaning of each character and the user's physical activities in a gamified environment. We verified the practicality of the platform through a user study investigating its overall usability and demonstrated it at a design exhibition to disseminate it to a wider audience.

The users perceived the system to be enjoyable, engaging, immersive, and helpful for learning logograms, while it did not cause motion sickness. Specifically, the users indicated that gamified actions through gestures and walking were the most fun part of the game, demonstrating the effectiveness of an embodiment strategy. Our study results revealed that the embodied learning approach via gamified VR is best suited for adolescents, indicating the system's applicability to students.

The work contributes to the field of embodied learning by introducing an enjoyable method of logogram edutainment. The novelty of the suggested technique is the exploitation of logograms that feature thousands of distinct words for the construction of learning content and a corresponding virtual reality environment. While it revealed the effectiveness of the embodied logogram learning among teenagers, it also evokes the necessity for additional aids for adults to learn to use the system without technical support. Additionally, the desirable playtime and quantity of characters, as well as preferred time per characters might be employed for future studies when researchers consider the discussions of the configurable task volume and the 'engagement time'. Potential future work involves a scenario design with more abstract words, more convenient methods of virtual reality locomotion for embodied learning, as well as more extensive user studies with diverse populations.

**Author Contributions:** Conceptualization, G.K.; methodology, G.K. and S.K.; software, E.A.; data curation, E.A.; formal analysis, G.K.; writing—original draft preparation, G.K. and E.A.; writing—review and editing, G.K. and S.K.; supervision, S.K. All authors have read and agreed to the published version of the manuscript.

**Funding:** This research was supported in part by the National Research Foundation of Korea (NRF) funded by the MSIT (2021R1A4A1030075), in part by the Korea Institute of Energy Technology Evaluation and Planning (KETEP) and the Ministry of Trade, Industry & Energy (MOTIE) of the Republic of Korea (No. 20204010600340), and in part by the IITP grant funded by the MSIT (No. 2019-0-01842, AI Graduate School Program (GIST)).

**Institutional Review Board Statement:** The study was conducted according to the guidelines of the Declaration of Helsinki, and approved by the Institutional Review Board of Gwangju Institute of Science and Technology. (Protocol code 20190510-HR-45-02-02, approved on 10 May 2019).

**Informed Consent Statement:** Informed consent was obtained from all subjects involved in the study.

**Data Availability Statement:** The data presented in this study are available on request from the corresponding author.

**Conflicts of Interest:** The authors declare no conflict of interest.

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
