# Peer review of "Logogram VR: Treadmill-Coupled VR with Word Reflective Content for Embodied Logogram Learning"

_applsci, doi:10.3390/app13031627_

Round 1
Reviewer 1 Report
CONFIDENTIAL COMMENTS FOR THE Authors:
1. Author(s) need to well- illustrate your viewpoints by relating your own methodology.
2. Figures can help readers to well-explain how the research work and provide international readership to get a better understanding of your research.
3. Update your references from 2012 to 2022
4. Make it more clear, please
Reviewer Strongly recommends 4 various types of articles for Suggested references:
Suggested references: for language learning Environments
1. English e-learning in the virtual classroom and the factors that influence ESL (English as a Second Language): Taiwanese citizens’ acceptance and use of the Modular Object-Oriented Dynamic Learning Environment
updated 1.4.22
Author Response
We appreciate reviewer 1 for the constructive feedback. Please find the file attached to track the revisions we made to reflect the comments.

Reviewer 2 Report
This paper presents a proof-of-concept solution for learning logograms using VR. The topic is interesting, shows novelty and it is well presented overall. My recommendation is acceptance after revision.
Detailed comments follow.
Line 101: there is one ) too much after [39]).
Lines 112-114:
I was first uncertain about what characteristics of the logogram will be mapped. Later, it was clear that both the meaning and the strikes play a significant role. It would be helpful to insert just one sentence here to tell the reader what is the plan: big gesture strikes, meaning, controlling the VR events etc. Furthermore, it was also not clear if one logogram is one scenario (that was my thought initially), but instead, you use one scenario with more logograms related.
You use a special treadmill. I haven’t seen and experienced this before, only the regular gym-like treadmills for running exercise (actually, we use a similar approach where these are connected to a VR environment). Probably, many readers are unfamiliar with this equipment. Maybe you could write some sentences about it (with a better picture?), how does it work, especially focusing on safety. We usually face problems using a VR HMD together with a (slow moving) treadmill, losing balance, falling etc. Is this not an issue here? Do subjects feel safe during the experiment?
Section 3.1.1. here you introduce the main three logograms. We don’t see them, and only one of them is partly visible later on a figure. I would include a picture here about these three characters in its usual written form subjects have to mimic later.
Lines 178-192: I felt this section a bit misplaced here, you may consider to move it into the introduction? Not necessarily, though.
Section 4:
It was not clear throughout the paper, whether this method is used and meant to be only for second language (L2) learning and/or also for native speaker children (L1). Similarly, the people at the conference were native speakers? Did you ask them about their prior knowledge of the symbols? I think this is a very important question, first of all, if real learning is involved. Older people, who may already know the logogram will have a different opinion. Furthermore, this influences very much the average time/character or time spent. If I am familiar already with the symbols, I will perform better and faster and I would not know how effective the teaching method is.
The experience and knowledge of the symbol is different from the experience about VR scenarios in general. I understand, you tested the latter, but large differences in avg. time and character number can be due to this as well.
You indicated some very important facts about this (line 406, 467), as the real test would be to test the actual learning effect, using the system for L2 only (or L1 for children) with no a-priori knowledge about the symbols. Using control groups who deal with a regular way of learning (with teacher and books) vs. VR system.
Reading the feedbacks, I totally agree to revise the necessity of the treadmill. Although it makes the VR experience better, I doubt that it is necessary for the learning, and the whole experiment could be repeated without it (just by looking around in the VR scenario, moving virtually with the controller if needed, even in a sitting position – line 386). This can enhance acceptance, it is cheaper, easier. Furthermore, maybe the VA can be also replaced by a human instructor prior the experiment or using written or spoken commands (line 333).
In our own experiments, we seldom detected sickness, dizziness for short periods (less than 10-15 minutes). On the other hand, longer sessions exceeding 20-25 minutes are more demanding and it happens more often. This seems to me some kind of time limitation, so you probably would have the same effect.
Line 340: the subjects have to speak and articulate the symbols. It was detected by a microphone. However, there is no information about it, how this was then used, evaluated. Did you make a comparison with existing speech samples? Does the VA give some kind of feedback? What if the pronunciation is good (or not)? I missed this part in the paper.
A final remark and note: as you have a relatively large variance in the avg. number of characters and time spent, it indicates that there will be users who are slow and can handle a small number of symbols, and others who can perform significantly better.
I would design scenarios without having a fix number of symbols, but having a list of about 10 related symbols and users may set the number individually. E.g., having different difficulty levels of “easy” (3 symbols), “moderate” (6 symbols) and “hard”. They can progress by finishing each level.
Author Response
We appreciate reviewer 2 for the constructive feedback. Please find the file attached to track the revisions we made to reflect the comments.

Reviewer 3 Report
Thank you for your interesting research,
First of all I am suffering from motion seek so you may not count me on your program... I have several questions in order to clarify your research purposes:
1. Is it really necessary to learn reading a symbolic language using VR?
2. Can you apply Log VR for learning syllable writing, eg. linear A & B?
3. How does affect the language of the students the performance of Log VR?
4. Can you apply Log VR for other symbolic writing systems?
5. If symbols are so connected to concepts, can you write Hanja by simply scan the brain activity through Log VR?
Congrats!
Author Response
We appreciate reviewer 3 for the constructive feedback. Please find the file attached to track the revisions we made to reflect the comments.
